# Method for the Detection of Tumor Blood Vessels in Neurosurgery Using a Gripping Force Feedback System

**DOI:** 10.3390/s19235157

**Published:** 2019-11-25

**Authors:** Hiroki Yokota, Takeshi Yoneyama, Tetsuyou Watanabe, Yasuo Sasagawa, Mitsutoshi Nakada

**Affiliations:** 1Institute of Science and Engineering, Kanazawa University, Kanazawa 9201192, Japan; yoneyama@se.kanazawa-u.ac.jp (T.Y.); twata@se.kanazawa-u.ac.jp (T.W.); 2Faculty of Medicine, Kanazawa University, Kanazawa 9200934, Japan; y-sasa@med.kanazawa-u.ac.jp (Y.S.); mnakada@med.kanazawa-u.ac.jp (M.N.)

**Keywords:** neurosurgery, surgical robot, blood vessel detection, cross-correlation coefficient

## Abstract

Avoiding unnecessary bleeding during neuroendoscopic surgeries is crucial because achieving hemostasis in a narrow operating space is challenging. However, when the location of a blood vessel in a tumor cannot be visually confirmed, unintentional damage to the vessel and subsequent bleeding may occur. This study proposes a method for tumor blood vessel detection using a master–slave surgical robot system equipped with a force sensor in the slave gripper. Using this method, blood pulsation inside a tumor was detected, displayed as a gripping force wave, via the slave force sensor. The characteristics of gripping force due to blood pulsation were extracted by measuring the fluctuation of the force in real time. The presence or absence of blood vessels was determined on the basis of cross-correlation coefficients between the gripping force fluctuation waveform due to blood pulsation and model fluctuation waveform. Experimental validation using two types of simulated tumors (soft: E = 6 kPa; hard: E = 38 kPa) and a simulated blood vessel (E = 1.9 MPa, radius = 0.5 mm, thickness = 0.1 mm) revealed that the presence of blood vessels could be detected while gripping at a constant angle and during transient gripping.

## 1. Introduction

For safe and efficient surgery, it is important to accurately understand the status of the operating area for preventing accidental damage. During endoscopic surgery, the characteristics of the operating area are determined on the basis of visual information obtained through the endoscope. However, when bleeding occurs, the surgical site is covered in blood, disrupting visualization. Particularly, during brain tumor resection under a neuroendoscope, the surgical site is often covered in blood even with minor bleeding because the operating space is very narrow. Therefore, frequent hemostasis is necessary, which places a great burden on the surgeon. The surgeon may intend to resect a tumor with minimal hemorrhage but can unintentionally cut a blood vessel in the tumor that could not be confirmed visually. Therefore, identification of the presence of blood vessels during brain tumor resection is critical.

A surgical robot is used to improve the safety and efficiency of surgery. In recent years, minimally invasive surgeries using a surgical robot have been frequently performed, particularly for urological and gynecological procedures [1,2,3]. Most surgical robots are master–slave robots, which allow for more dexterous movements through scaling and greater degree of freedom for surgeries in humans. Due to these merits, various surgical robots have been developed, particularly for laparoscopic surgeries [4,5]. However, most of these master–slave surgical robots lack a force feedback system. Force sensing plays an important role in determining the status of the operative area, and lack of force information can lead to tissue damage [6,7]. Therefore, various methods have been evaluated to estimate the working force exerted by the operating segment in surgeries using a master–slave robotic system [8,9,10,11]. Additionally, some studies have used a force/tactile feedback system of surgical robots to enable the operator to palpate the tissue at the operative site through which they can detect blood vessels in the region of interest [12,13].

In addition to palpation using force/tactile feedback, many other blood vessel detection methods have been investigated. Peine et al. [14] developed a probe with a tactile sensor at the tip, which successfully detected pulsation of the radial artery. Beasley et al. [15] developed a blood vessel tracking system using tactile sensors, which detected the position of a blood vessel under the tissue based on the intensity of pulsation in each array of tactile sensors. In addition, there are various methods for detecting blood vessels beyond the use of tactile sensors. For instance, Chaturvedi et al. [16] developed a laparoscopic grasper, which can detect and visualize subcutaneous blood vessels by incorporating a near infrared LED and a linear imaging sensor into the opposing jaws of the surgical grasper. Kim et al. [17] developed a sensor to detect changes in the impedance of the vascular wall resulting from the expansion and contraction of blood vessels using a constant current source and a voltage sensor; using this sensor, blood vessels with an inner diameter of 4–18 mm could be detected. McKinley et al. [18] developed a palpation probe using Hall effect sensors. This probe is made from 3D-printed and commercially available electronic parts and is disposable. Experiments using phantoms suggested that a subcutaneous blood vessel phantom with a maximum depth of 5 mm and diameter of 2.25 mm was detected. Akbari et al. [19] developed a method to detect blood vessels and distinguish arteries from veins through hyperspectral imaging via a visible light camera and two infrared cameras with different wavelength ranges. Park et al. [20] developed an instrument capable of imaging blood vessels, which incorporated the optical frequency domain imaging technology into a laparoscopic instrument. The instrument succeeded in observing blood vessels with a diameter of 2.2 mm in an experimental pig model. These sensors and tools for detecting blood vessels are mainly used for laparoscopic surgery. Blood vessels can be detected by applying these sensors and tools to the operating segment of laparoscopic surgical robots. However, neurosurgery is challenging because the operating space in this surgery is very narrow compared with that in laparoscopic surgery. Thus, it is difficult to use these sensors and tools in neurosurgery.

Ramakonar et al. [21] developed an “imaging needle” to monitor blood vessels during surgery by attaching an optical coherence tomography sensor to a biopsy needle and succeeded in detecting blood vessels in vivo. To use this system in an existing neurosurgical robotic system, it is necessary to attach another sensor to the slave manipulator. However, the slave manipulator used for neurosurgery is very small and narrow, and there is no extra space to add another sensor. For safe and efficient surgery, both a force feedback system and a blood vessel detection system are necessary. Using both, the operator can better determine the status of the tissue, which would improve the safety and efficiency of the operation.

In this article, we propose a blood vessel detection method for neurosurgery. This method uses a system of force feedback with a surgical robot. The authors previously developed a surgical robot manipulator with a force feedback system and demonstrated the benefits of the system in neurosurgery [22,23]. However, the blood vessel detection ability of this system is yet to be verified. In laparoscopic surgery, blood vessels can be detected using a force/tactile feedback system. However, in neurosurgery, it is difficult to detect blood vessels using the existing force/tactile feedback system because the thickness of the target blood vessels is different from that in laparoscopic surgery. Specifically, the target blood vessels in laparoscopic surgery are thicker than those in neurosurgery and can be easily detected with the force/tactile feedback system, while in neurosurgery, the target blood vessels are very small. Therefore, it is difficult to determine the mechanical compliance of the blood vessels, even using the force feedback system.

In this study, with the aim of preventing bleeding caused by unintended damage to tumor blood vessels, we developed a novel blood vessel detection method that uses a master–slave surgical robot system with a force sensor at the operating segment. The method detects blood pulsation based on gripping force. The proposed method was verified through a gripping experiment using a neurosurgical robot manipulator with a force feedback system developed by our research group. Simulated tumors and blood vessels were used in the gripping experiments. Pulsation of the simulated blood vessels inside the simulated tumors could be detected using the slave gripper force sensor.

## 2. Materials and Methods

### 2.1. System Design

Figure 1 shows simulated blood vessel detection during tumor resection. If a robotic gripper does not detect a blood vessel inside the tumor, then bleeding may occur upon pulling of the gripped tumor. If the gripper detects the blood vessel while gripping the tumor, however, then pulling the tumor can be avoided and bleeding can thus be prevented. Therefore, this study aimed to detect the presence of blood vessels via gripped tumors. The system is designed to detect arteries but not veins and capillaries. In addition, blood vessels that adhere to the tumor surface, or are otherwise visually apparent, are not targeted.

To develop a method that senses fluctuations in the gripping force of a tumor caused by blood pulsation, thereby identifying embedded blood vessels, the following factors were considered:(1)Dependence on tumor hardness. Some tumors are hard and some soft, and differences in tumor hardness may affect the gripping and blood pulsation forces.(2)Extraction method of the pulsation waveform. Force changes due to blood pulsation must be considerably lesser than the gripping force. A method to remove the gripping force and extract the pulsation waveform alone must be developed.(3)Blood pulsation detected by changing the gripping force. While the blood pulsation waveform is approximately 1 Hz, the gripping force fluctuation varies based on the gripper operation. A method to distinguish the blood pulse waveform from the gripping force fluctuation waveform must be established.

A detection system was configured using a neurosurgical robotic system based on the abovementioned factors (Figure 2). In accordance with (1), tumor model materials used were urethane gel and silicone rubber. These materials have different mechanical compliances. Regarding (2), detection performance at a constant gripping angle and gripping force was evaluated to determine the gripping force necessary to detect vascular pulsation. Regarding (3), detection performance was investigated during gripping with increasing force.

Figure 2a,b show the detection system at a constant gripping angle and during the gripping process, respectively. Both detection systems comprise three parts: detection of the gripping force using a gripper equipped with a force sensor, extraction of blood pulsation from the gripping force, and blood pulse analysis. In the constant grip angle experiment, the frequency analysis was performed after extracting the blood pulse characteristics at different gripping angles. In the increasing gripping force experiment, curve-fitting inspection was performed to detect blood pulse. Curve-fitting was applied to quickly detect blood pulse while changing the gripping force. Cross-correlation coefficients between the gripping force and model fluctuation waveforms were calculated. Time discrimination in the curve-fitting analysis is expected to be shorter than that in the frequency analysis.

### 2.2. Force-Detecting Gripper

The robotic system is a master–slave system that uses bilateral control, in which the slave manipulator moves following the master manipulator and provides feedback for the force detected by the slave manipulator to the master manipulator. Force is detected by a gripper with a force sensor at the tip of the slave manipulator (Figure 3). The force sensor has a parallel plate structure and uses strain gauges (Kyowa Electric Industry, KFRS-02-120-C1-N50C2) to detect the gripping and pulling forces. Force in each direction is measured by a bridge circuit using four strain gauges: two active gauges that measure bending strain and two dummy gauges for temperature compensation. This is because the gripper is very small and there is no space to place four strain gauges for each type of force. Resistance of the strain gauge changes due to temperature changes in the measurement environment. The resistance change due to temperature change is measured by two strain gauges that are not subjected to bending strain. By correcting the resistance change measured by the two dummy strain gauges in the bridge circuit, the resistance change of the two active strain gauges, which are subjected to the bending strain, can be accurately measured. The parallel plate structure allows for the detection of bending strain in the gripping force direction without interference from other directional forces when force is applied to the tip of the gripper. The output of the gripping force sensor is 1V/1N from the amplifier, and the 16-bit A/D convertor accepts the input range of ±10 V. Sampling time is 5 ms. The manipulator that detects the gripping force is described elsewhere [22].

Blood vessel pulsation is detected as a function of the gripping force. The tumor is compressed by a gripper equipped with a force sensor. Compression of the tumor facilitates the inflation of blood vessels inside the tumor to the surface. Understanding the tumor’s mechanical compliance using force sensing and applying appropriate compression produces a gripping force fluctuation waveform.

### 2.3. Extraction of Blood Pulsation Characteristics

The gripping force data detected by this system include the blood vessel and tumor gripping forces in the tumor to be removed. Trend data for the gripping force, which appears as a low-frequency component in signal processing, should be removed to extract the pulsating waveform. Typically, trend removal is performed after the targeted waveform appears. However, a time delay occurs in real-time processing. In this system, quick detection is necessary to avoid interference during the surgery. Therefore, instead of trend removal, the blood pulse is extracted by measuring the fluctuation of the gripping force that occurs during a single blood pulse. For the measurement of fluctuation, a second-order approximate value of gripping force is used. This value is calculated using data size for a single pulsation to obtain average fluctuation of gripping force during a single pulsation. By subtracting the second-order approximate value of the gripping force from the actually measured gripping force value, the fluctuation occurring during a single pulsation can be calculated. This method has no time delay.

The sampling frequency of the system must be sufficient to measure the range of pulsation. In this article, the maximum frequency of pulsation is assumed to be approximately 1.7 Hz; thus, a sampling frequency of 3.4 Hz or more is required. In addition, to extract fluctuations that occur during a single pulsation, the calculation of the second-order approximate value requires a data size that can include a single pulsation. In this neurosurgical robotic system, the sampling frequency is 200 Hz, and the target pulsation range is 1.0–1.7 Hz. The data size used for the calculation was 1 s (200 samplings). Now, the second-order approximation value ${f^{\prime }}_{0}$ is given by
(1)$${f^{\prime }}_{0}=at_{0}{}^{2}+bt_{0}+c$$
where the second-order approximation coefficients a, b, c are given by
(2)$$\left[\begin{array}{c} a\\ b\\ c \end{array}\right]=\left[\begin{array}{ccc} \sum t_{i}{}^{4} & \sum t_{i}{}^{3} & \sum t_{i}{}^{2}\\ \sum t_{i}{}^{3} & \sum t_{i}{}^{2} & \sum t_{i}\\ \sum t_{i}{}^{2} & \sum t_{i} & \sum 1 \end{array}\right]\left\lceil \begin{array}{c} \sum t_{i}{}^{2}f_{i}\\ \sum t_{i}f_{i}\\ \sum f_{i} \end{array}\right\rceil \;(i\;=\;0\;\sim \;199)$$
where t is the measured time data, f is the measured gripping force data, and i is the data index from the frame data. A data index of zero represents the frame data. Since the sampling time for this system is 5 ms, a data index of 199 represents the data 1 s earlier. Next, the second-order approximation value is subtracted from the frame data of gripping force. Gripping force fluctuation data $d_{0}$ is given by
(3)$$d_{0}=f_{0}-{f^{\prime }}_{0}$$

A schematic diagram for the extraction of blood pulse in this system is shown in Figure 4. In this method, a sine waveform is generated from one blood pulse. The magnitude relationship between the actually measured gripping force and the second-order approximate value varies depending on whether the blood pulse waveform generated on the measured gripping force is rising or falling. Therefore, when the second-order approximate value is subtracted from the measured gripping force, the section where the blood pulse waveform is rising and the section where the blood pulse waveform is falling are generated as a sine waveform.

### 2.4. Discrimination of Blood Pulse at a Constant Gripping Angle

Blood pulse rates range from 60 to 100 bpm, depending on the individual and their physical condition, and the average value is 72 bpm for an adult [24]. In this article, assuming the pulse rate is 72 bpm, a waveform of 1.2 Hz for the gripping force data is defined as the pulsation signal. Frequency analysis using Fourier transform was used to determine the magnitude of the blood pulse components.

### 2.5. Discrimination of Blood Pulse during the Gripping Process

When the gripping angle is constant, the pulse can be accurately detected and the blood vessel can be identified based on the frequency analysis. However, in the gripping process, the increasing speed of the gripping force depends on the gripping speed. The frequency of the gripping motion may be faster than the blood pulse frequency. In such a case, it is difficult to identify the pulse by frequency analysis. Therefore cross-correlation analysis is applied. The cross-correlation coefficient is calculated by curve-fitting between the gripping force fluctuation waveform and the model fluctuation waveform. A schematic diagram for curve-fitting is shown in Figure 5.

Curve-fitting is performed as follows: to calculate cross-correlation coefficients, the gripping force fluctuation model waveform for one cycle is used. The cross-correlation coefficient is defined by normalizing the cross-correlation function between the two fluctuation waveforms with auto correlation functions. The cross-correlation coefficient $\rho_{xy}$ is given by
(4)$$\rho_{xy}=\frac{\varphi_{xy}\left(\tau \right)}{\sqrt{\varphi_{xx}\left(0\right)\varphi_{yy}\left(0\right)}}$$
where $\varphi_{xy}\left(\tau \right)$ is the cross-correlation function between the gripping force fluctuation waveform and the model fluctuation waveform at the time lag τ, $\varphi_{xx}\left(0\right)$ is the autocorrelation function of the gripping force fluctuation waveform at the time lag τ = 0, and $\varphi_{yy}\left(0\right)$ is the autocorrelation function of the model fluctuation waveform at the time lag τ = 0. The cross-correlation coefficient is calculated while changing the value of lag τ of $\varphi_{xy}\left(\tau \right)$, and the value is acquired when the highest correlation coefficient is acquired. After calculation, gripping force fluctuation data are newly acquired, and one sample of the gripping force fluctuation data used for calculation is updated. Then, cross-correlation coefficients with the model fluctuation waveform were calculated again. In the robotic system, these processes are performed in real time.

## 3. Experimental System

Experimental setup is shown in Figure 6. Urethane gel (EXSEAL, Inc. HITOHADA clear type) and silicone rubber (Smooth-On, Inc. Ecoflex 00-10) were used to simulate tumors, and a silicone rubber tube (ARAM, radius = 0.5 mm, thickness = 0.1 mm) was used to simulate a blood vessel. These materials were selected based on mechanical compliance. The dynamic viscoelasticity of urethane gel, with a10% salt added by weight, is close to that of pig brain [25], making urethane gel appropriate for simulating a tumor. The mechanical elasticity of brain tumors varies depending on the site where they occurs and is around 11.4 ± 3.6–33.1 ± 5.9 kPa [26]. The Young’s moduli of urethane gel and silicone rubber, measured in a preliminary experiment, were 6 and 38 kPa, respectively.

The Young’s modulus of the cerebral artery varies depending on its diameter and wall thickness. In a study of fluid analysis in cerebral aneurysm, the Young’s modulus of a normal blood vessel wall in the brain was found to be 1 MPa [27,28]. Based on these results, silicone rubber tubing was selected to simulate a blood vessel. The Young’s modulus of the silicone rubber tube measured in a preliminary experiment was 1.9 MPa. In this experiment, a combination of a simulated tumor and blood vessel was used as the gripping object. Both urethane gel and silicone rubber can be prepared by mixing a primary liquid agent with a liquid hardening agent. The gripping object was produced by passing a silicone rubber tube through urethane gel in a liquid state and curing.

During fabrication of the urethane gel, 10% salt (by weight of the mixed liquid) was added to the mixed liquid. Additionally, the surface of the urethane gel after curing was viscous, and adhesion of the urethane gel to the gripper caused a disturbance in the measurement of gripping behavior. Therefore, talc powder was applied to the surface of the urethane gel to reduce adhesion. In addition, pressure fluctuation was applied to the silicone rubber tube in the gripping object. This simulated the inflation of the blood vessel due to blood pulsation.

The device used to apply internal pressure fluctuation to the silicone rubber tube is shown in Figure 7. The bellows of the device were filled with air and connected to the silicone rubber tube via the urethane tube and a syringe. Water was added from the urethane tube to the silicone rubber tube. The silicone rubber tube was inflated when the water inside was pressurized by air from the bellows. The pressure in the silicone rubber tube was measured using a pressure sensor (SENSEZ, Inc. HTV_050KP_V) attached in the direction opposite the silicone rubber tube through the joint. A small artery was targeted for detection in this experiment. The systolic blood pressure for a small artery is approximately 110 mmHg (14.6 kPa) and the diastolic pressure is 70 mmHg (9.3 kPa) [29]. The pulse pressure, which is the difference between the diastolic and systolic blood pressures, is ~5.3 kPa. In this experiment, inflation of the blood vessel was reproduced. Considering the Young’s modulus of the silicone rubber tube, internal pressure fluctuation was applied to the silicone rubber tube at 11 kPa, which is nearly two times the actual pulse pressure.

## 4. Blood Pulse Detection at a Constant Grip Angle

### 4.1. Testing Method

To evaluate the detection performance of the system, the effect of the pulsation signal on the gripping force data at various gripping angles was measured. In this experiment, the slave gripper was driven by directly controlling the actuator of the gripping motion. The schematic view of the experimental workflow is shown in Figure 8. The initial state of the gripper was opened at 34°, defined as a gripping angle of 0°, and in the next stage, the closing angle reduced from the initial angle was defined as the gripping angle. The gripping speed was constant, regardless of the target gripping angle. Six gripping angles were selected as targets. Regarding the gripping object, two different simulated tumors made of urethane gel and a silicone rubber were used. For both gripping objects, the silicone rubber tube inside the simulated tumor was pressurized at 1.2 Hz. The strength of the pulsation signal on the gripping force data at each gripping angle was evaluated based on the amplitude spectrum. Fast Fourier transform (FFT) was used to calculate the amplitude spectrum. For FFT, the gripping force fluctuation data were used.

### 4.2. Test Results

The gripping force data detected by the gripper force sensor at each target gripping angle are shown in Figure 9a. Using the 30° gripping angle with a gripping force of 0.5 N, a minute periodic waveform appeared on the gripping force between 17 and 45 s. The gripping force fluctuations detected at each target gripping force (by subtracting the second-order approximate value) are shown in Figure 9b. The amplitude of fluctuation waveform was markedly high at the 30° angle.

The gripping force data for the silicone rubber tube detected using the gripper force sensor at each target gripping angle are shown in Figure 10a. At 20° and 25° gripping angles with gripping forces of 0.4 and 0.7 N, respectively, the fluctuation wave appeared in the gripping force. However, the wave disappeared at the 30° gripping angle with a gripping force of 1.1 N. The gripping force fluctuation for the silicone rubber is shown in Figure 10b. The amplitude of fluctuation waveforms was markedly high at 15°–25° gripping angles but low at 30°.

FFT was performed to determine whether the gripping force fluctuations were pulsation waveforms and to collect the gripping force fluctuation data at a constant gripping angle interval (window size, 4096; sampling interval, 5 ms; 20–40 s). The amplitude spectrum for the 1.2-Hz component was selected as the amplitude spectrum for the pulsation signal because the silicone rubber tube was pressurized at 1.2 Hz in this experiment. The amplitude spectrum for urethane gel detected at each target gripping angle compared with the baseline without pressurization of the silicone rubber tube is shown in Figure 11a. Error bars represent standard deviation. The amplitude spectrum was markedly high at the 30° gripping angle.

The amplitude spectra for silicone rubber detected at each target gripping angle are shown in Figure 11b, which increased at gipping angles from 15°–25° but decreased at 30°.

Table 1 shows the results of paired t-tests. Moreover, t-tests (two-sided) were performed for a pair of samples with the “pulse pressure” and “baseline” amplitude spectra. In addition, t-tests were conducted on the distributions of pulse amplitude spectra for 1.2 Hz between the gripping force data with and without pulse pressure in triplicate at each gripping angles (Figure 11).

When using the urethane gel, a difference was found at a significant level of 5% between the “pulse pressure” and “baseline” amplitude spectra at the 30° gripping angle alone. When using the silicone rubber, however, differences were found at a significance level of 5% between the “pulse pressure” and “baseline” amplitude spectra at 15°, 20°, and 25° gripping angles.

Therefore, the gripping angle that could detect the blood pulse for the urethane gel was approximately 30° and that for the silicone rubber tumor was 15°–25°.

## 5. Detection of Blood Pulsation in the Gripping Process

### 5.1. Testing Method

This section describes the experiments to detect blood vessel pulsations while the surgical robot is in the process of gripping a tumor using a manipulator. The gripping process is defined as the period from when the gripper starts closing until a constant gripping angle is reached. In this experiment, the gripper was operated at various gripping speeds by directly controlling the actuator for the gripping motion. A schematic view of the experimental workflow is shown in Figure 12. The initial state was similar to the experiment described in Section 4. In this experiment, three gripping speeds were set until the target gripping angle was reached. Detection of blood pulsation in the gripping process was evaluated by calculating the cross-correlation coefficients using curve-fitting with the gripping force fluctuation during the period of a single pulse time just before the present time and model fluctuation waveforms shifted in this period. Then the maximum cross-correlation coefficient was recorded as the present value. The model fluctuation waveform used for curve-fitting is shown in Figure 13. Model fluctuation waveforms were different for urethane gel and silicone rubber. The 10-cycle waveforms of the gripping force fluctuation detected at a constant gripping angle were averaged, and a waveform that approximated the first-order sine wave was defined as the model fluctuation waveform.

The experiment was performed three times with pulse pressure and three times without pulse pressure as baselines for each gripping speed. To improve detection accuracy, subtraction with the spectrum of baseline waveform was performed on the measured gripping force fluctuation waveform before curve-fitting. The purpose of this process was to remove fluctuation components other than blood pulsation from the gripping force fluctuation waveform with the pulse pressure. The baseline spectrum amplitude was subtracted from the fluctuation waveform value with pulse pressure. The subtracted spectrum was inverse Fourier transformed for measuring gripping force wave.

### 5.2. Test Results

Gripping force behaviors using the urethane gel are shown in Figure 14, Figure 15 and Figure 16 at gripping speeds of 2.5°/s, 5.0°/s, and 10.0°/s, respectively. In each figure, the gripping force increased until the gripper stopped at 30° and then remained constant (Figure 14a, Figure 15a and Figure 16a). The grip force fluctuation waveforms, determined by subtracting the second-order approximate values, are shown in Figure 14b, Figure 15b and Figure 16b. The cross-correlation coefficients obtained through curve-fitting between the gripping force fluctuation and model fluctuation waveforms are shown in Figure 14c, Figure 15c and Figure 16c, where the vertical axes represent the cross-correlation coefficients, the lower horizontal axes represent time, and the upper horizontal axes represent the grip angle.

The periodic waveform caused by pulse pressure appeared in the gripping force fluctuation from 15 s in Figure 14b, and the cross-correlation coefficient remained high from 14 s in Figure 14c. This time is considered the pulse-detected time because the high value was maintained from this time onward, except reduction at around 16 s, which was difficult to remove the baseline because various amplitudes of frequency emerged when the gripper speed was reduced for stopping.

Similarly, periodic waveform caused by pulse pressure appeared from 9 s in Figure 15b, and pulse detection appeared at 8.5 s as the start of continuous high cross-correlation coefficient in Figure 15c.

At the gripping speed of 10.0°/s, periodic waveform caused by pulse pressure appears at around 7 s, just before large fluctuation of gripping force at the gripper stopping time in Figure 16b. Pulse detection time for the cross-correlation coefficient in Figure 16c was not clear while the gripper is closing.

Gripping force behaviors using the silicon rubber are shown in Figure 17, Figure 18, and Figure 19 at gripping speeds of 2.5°/s, 5.0°/s, and 10.0°/s, respectively.

Periodic waveforms caused by pulse pressure are shown in gripping force fluctuations from 12 to 17 s in Figure 17b. Periodic waveform disappeared after 17 s because the pulsation disappeared when the gripping angle reaches 30° in the case of silicone rubber, as shown in Section 4. Moreover, the cross-correlation coefficient remained high from 12 to 16 s (Figure 17c). This period was considered as the pulse-detected time.

Similarly, the periodic waveform caused by pulse pressure during gripping force fluctuation from 9 to 11 s is shown in Figure 18b, and the cross-correlation coefficient remained high from 9 to 11 s (Figure 18c). This period was considered as the pulse-detected time.

The periodic waveform caused pulse pressure appeared in the gripping force fluctuation from 6.5 to 7.5 s (Figure 19b), and the cross-correlation coefficient was high at around 7 s (Figure 19c). This period was considered the pulse-detected time.

## 6. Discussion

### 6.1. Blood Pulse Detection Performance

From the results of experiments conducted at a constant gripping angle, the detection performance appeared to depend on the Young’s modulus of the tumor.

If the tumor was soft, pulse detection was only available at the gripping angle of 30°, indicating that the possible pulse conduction distance is very small for soft tumors. When the tumor was hard (silicone rubber), possible pulse detection distance was large at gripping angles from 15° to 25°. Therefore, possible pulse detection distance increased as the tumor hardness increased.

In the detection experiments conducted during the gripping process, periodic gripping force fluctuations appeared at a certain angle during the increase of gripping angle for both the urethane gel and silicone rubber samples. Moreover, the appearance could be detected slightly earlier in the cross-correlation coefficient investigation than in the gripping force fluctuation wave.

Curve-fitting was useful to detect the pulse waves during the gripping process perhaps because of the subtraction of the gripping force without pulse from the gripping force with pulse.

The behavior of the cross-correlation coefficient depends on the gripping speed. Considering the influence of the gripping speed on the value of the pulse wave, since the value of pulse wave increases with increase in gripping force, increase in the pulse value wave also depends on gripping speed. For more precise curve-fitting, increasing the pulse wave model according to the gripping speed would be ideal for the analysis of cross-correlation coefficient.

If the gripping process is not sufficiently longer than the pulsation period, the waveform cannot be detected stably over one pulsation cycle; thus, the accuracy of curve-fitting decrease and high correlation coefficient cannot be obtained. In such a case, curve-fitting with a part of model pulse wave in various periods will be necessary to achieve rapid detection of the pulse.

### 6.2. Practicality of the Proposed System

Although we only present basic results on the detection of the blood pulse inside simulated tumors during the gripping process, we believe that this new system will be useful for surgeons operating the master manipulator. If the surgeon suspects the existence of blood vessels inside a tumor, then gripping the tumor can be prolonged to determine the blood pulsation in the gripping force wave of the slave manipulator. Otherwise, the tumor can be slowly gripped to detect the blood pulse in the gripping force.

The existence of blood vessels can be communicated to the surgeon in many ways, such as a visual display of the gripping force wave and cross-correlation coefficient value or alarm sounds based on the cross-correlation coefficient value.

Besides the simple gripping of a tumor, many other surgical situations should be considered. Ideally, the blood vessels inside a tumor should be detected when the gripper touches or approaches the blood vessel in the tumor from various directions. Even in such cases, the force-detecting gripper can detect any type of force, indicating the existence of blood vessels. Further investigation using various cases in approaching the blood vessel is necessary in future studies.

The method proposed here is useful for gripping the tumor and for determining the resection approach route to reach the tumor as hemorrhage can limit visibility and cause surgical delays.

## 7. Conclusions

This article proposes a blood vessel detection method for neurosurgery using a master–slave manipulator for tumor resection. In this method, blood vessels are detected using force-sensing grippers in the slave manipulator, and blood vessels that pass through a tumor are targeted for detection. While gripping a tumor that contains blood vessels, blood pulsation is detected using the gripping force wave of the gripper. The pulse is detected by observing the gripping force fluctuation wave and cross-correlation coefficient analysis. Curve-fitting is performed based on the gripping force fluctuation and model fluctuation waveforms to obtain cross-correlation coefficients. The proposed method was verified with gripping experiments using simulated tumors and blood vessels. Pulsation of a simulated blood vessel was detected with gripping at a constant angle and during the process of gripping.

Although the detection accuracy and speed of this method should be improved, this method will be useful to avoid hemorrhage during tumor resection and while making a route for tumor resection.

## Figures and Tables

**Figure 1 sensors-19-05157-f001:**
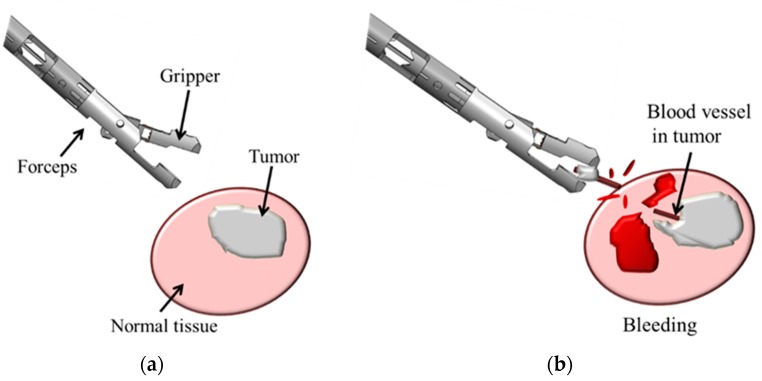
Importance of blood vessel detection during tumor resection. (**a**) No blood vessels are apparent on the target tumor. (**b**) Rupture and bleeding from the hidden blood vessel occur upon pulling the tumor.

**Figure 2 sensors-19-05157-f002:**
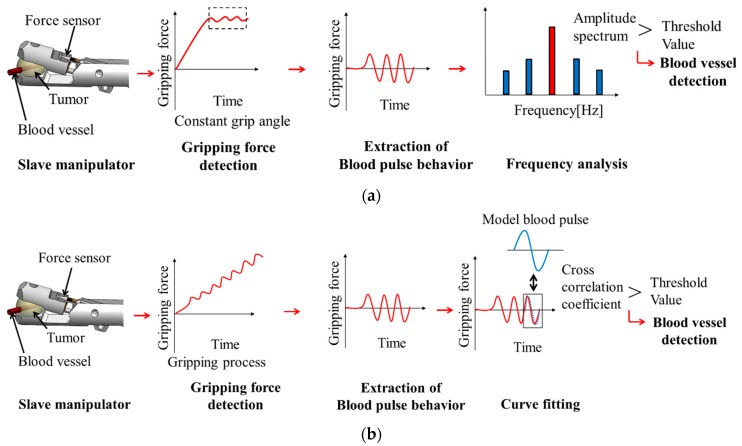
Blood pulse detection. (**a**) Detection at a constant gripping angle. (**b**) Detection during the gripping process.

**Figure 3 sensors-19-05157-f003:**
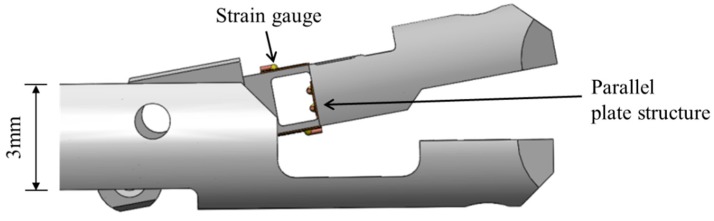
Force-detecting gripper.

**Figure 4 sensors-19-05157-f004:**
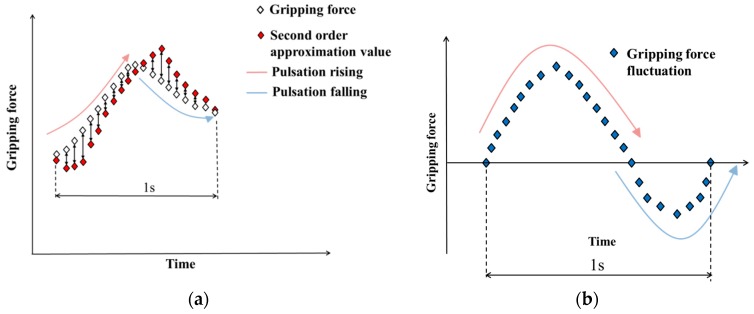
Extraction of pulse waveform using a second-order approximation value. (**a**) Gripping force data and second-order approximation value. (**b**) Extracted pulse wave.

**Figure 5 sensors-19-05157-f005:**
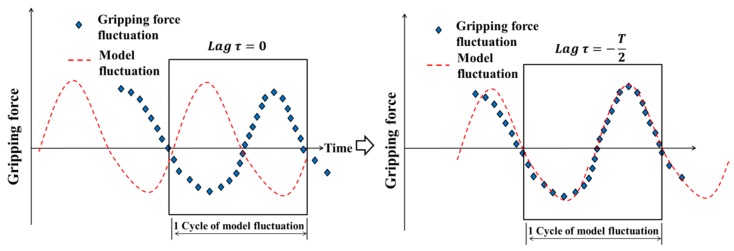
Schematic diagram of curve-fitting.

**Figure 6 sensors-19-05157-f006:**
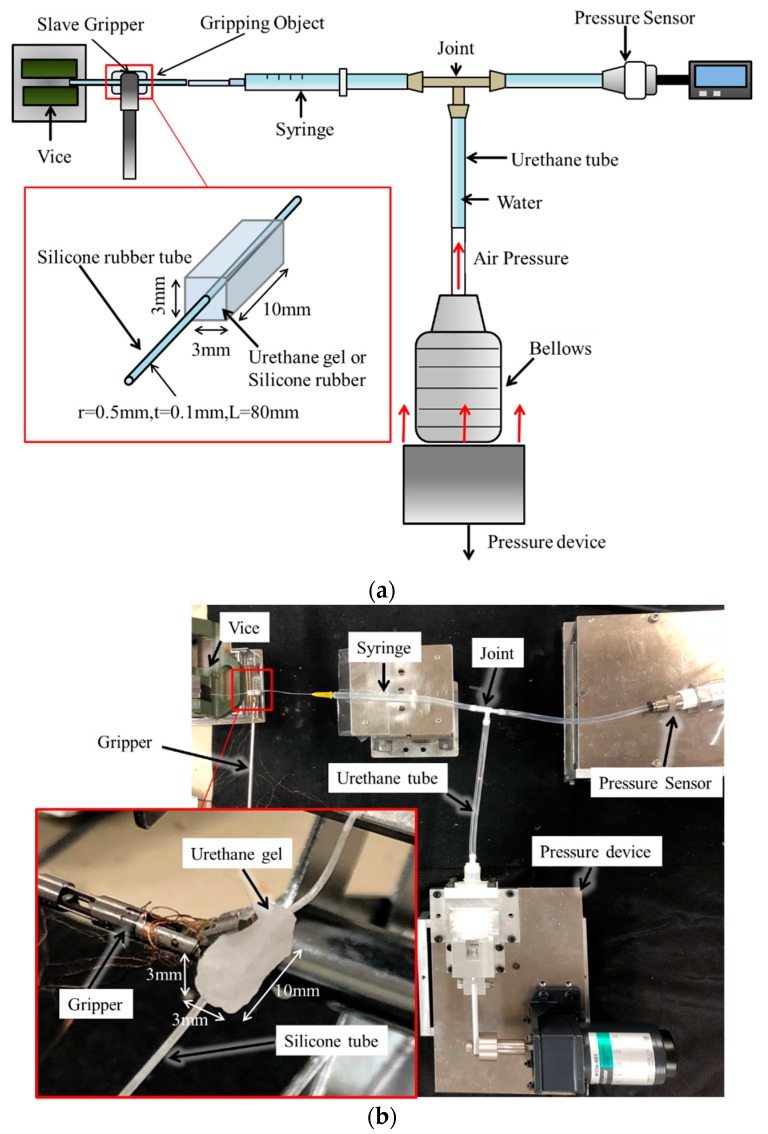
Experimental equipment. (**a**) Schematic view. (**b**) Picture (e.g., gripping object is urethane gel and silicone rubber tube).

**Figure 7 sensors-19-05157-f007:**
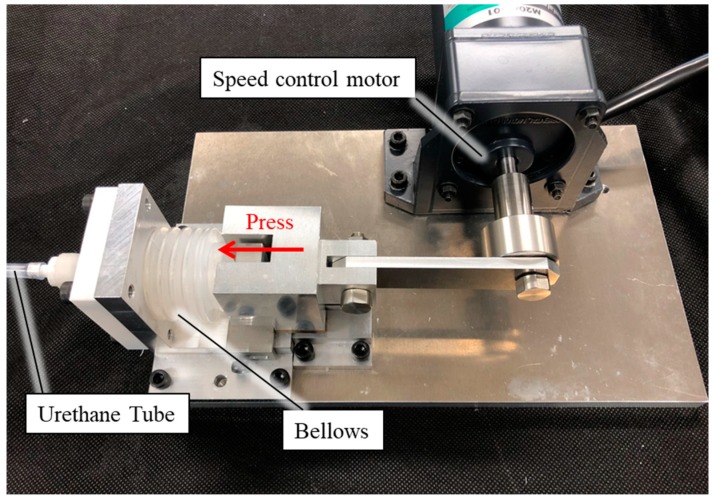
Pressure application device.

**Figure 8 sensors-19-05157-f008:**
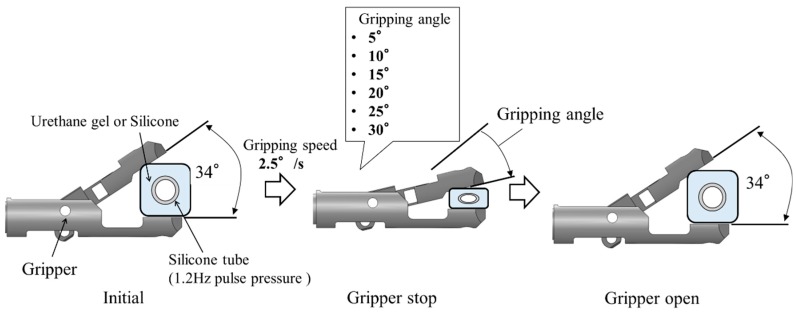
Schematic view of the experimental workflow.

**Figure 9 sensors-19-05157-f009:**
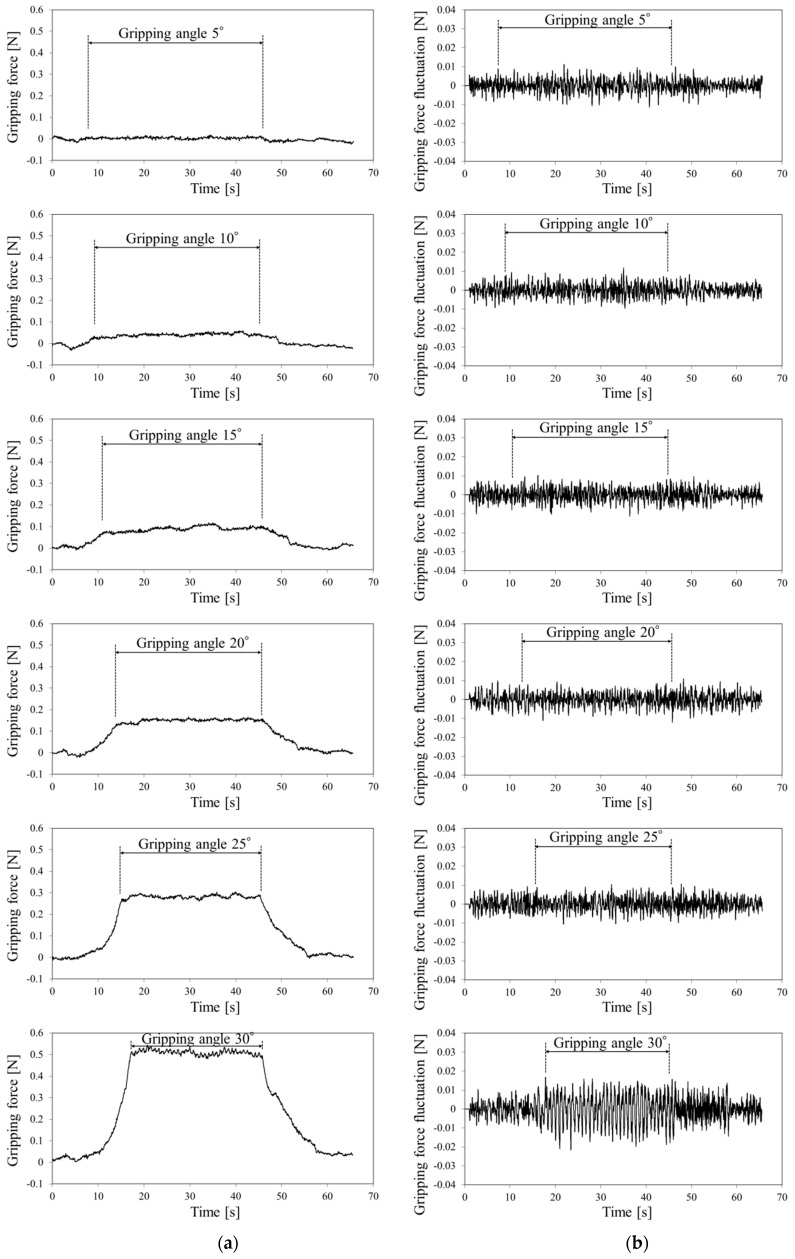
(**a**) Gripping force and (**b**) gripping force fluctuation for urethane gel.

**Figure 10 sensors-19-05157-f010:**
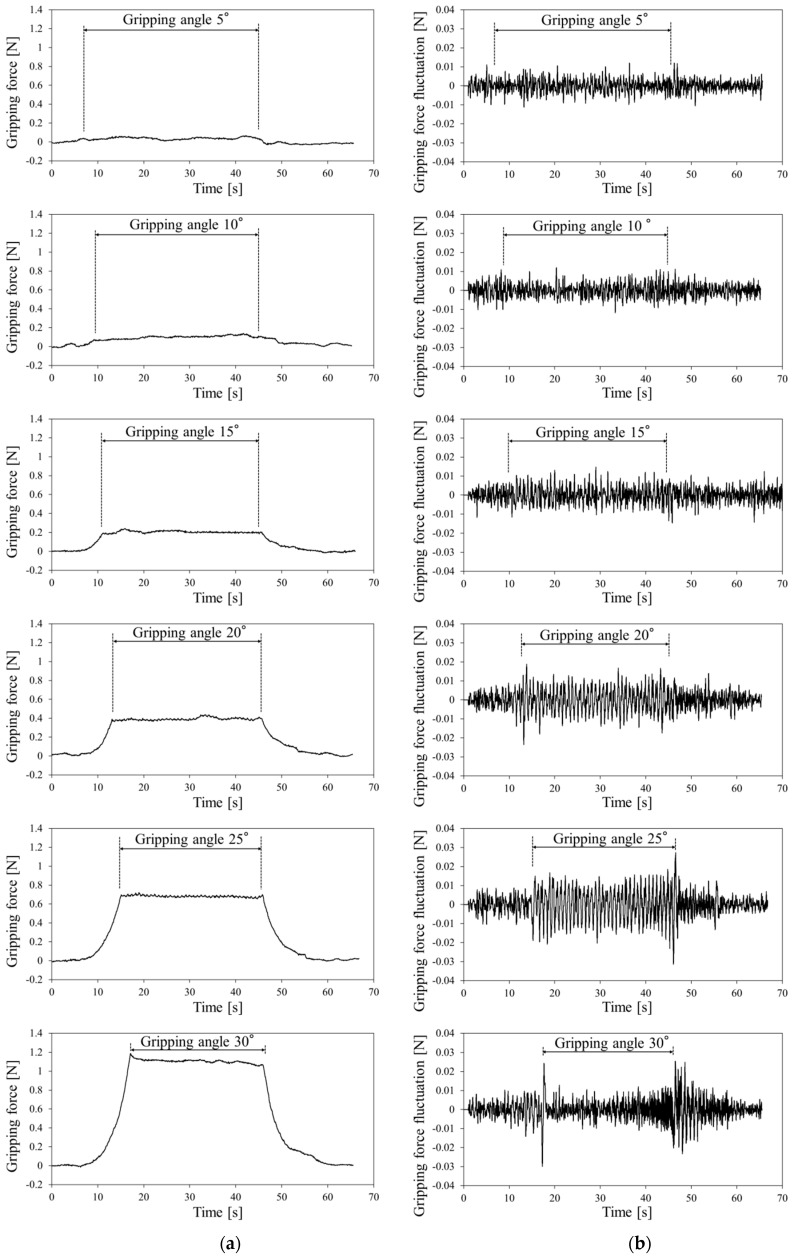
(**a**) Gripping force and (**b**) gripping force fluctuation for silicone rubber.

**Figure 11 sensors-19-05157-f011:**
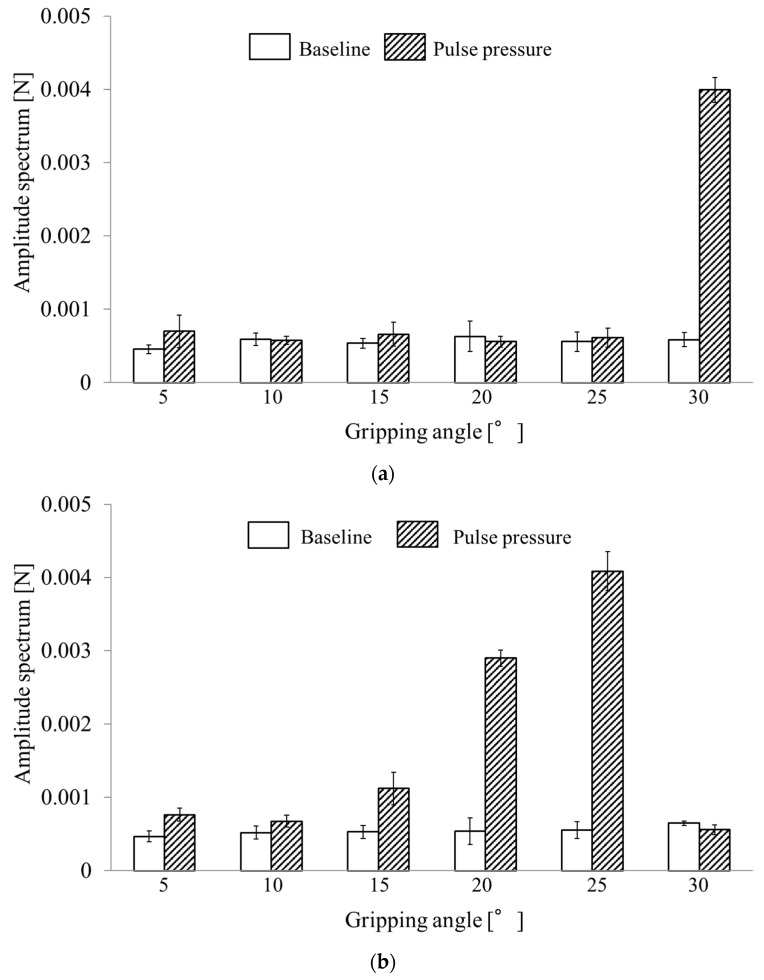
Pulse amplitude spectrum at each grip angle for (**a**) urethane gel and (**b**) silicone rubber.

**Figure 12 sensors-19-05157-f012:**
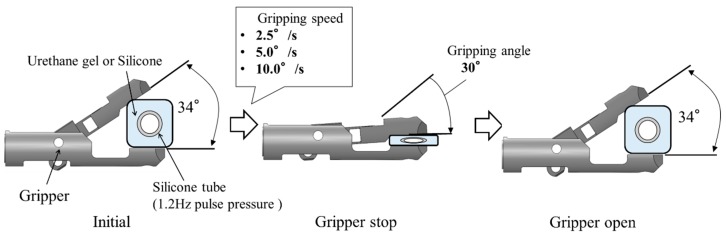
Schematic view of experimental flow.

**Figure 13 sensors-19-05157-f013:**
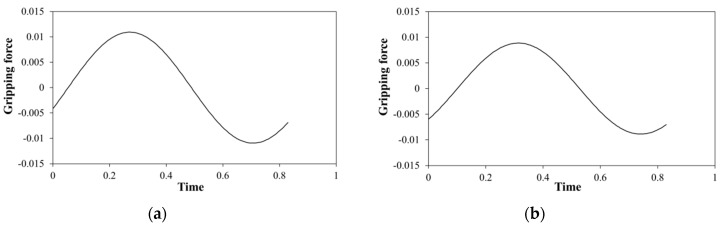
Model fluctuations of curve-fitting for (**a**) urethane gel and (**b**) silicone rubber.

**Figure 14 sensors-19-05157-f014:**
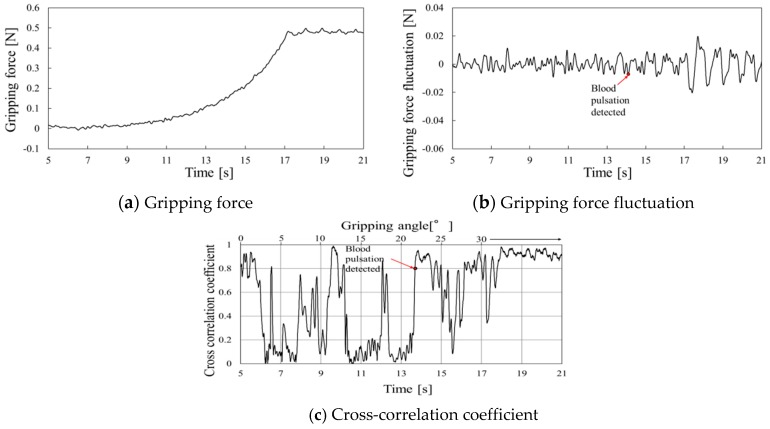
Gripping force behavior for urethane gel at a gripping speed of 2.5°/s.

**Figure 15 sensors-19-05157-f015:**
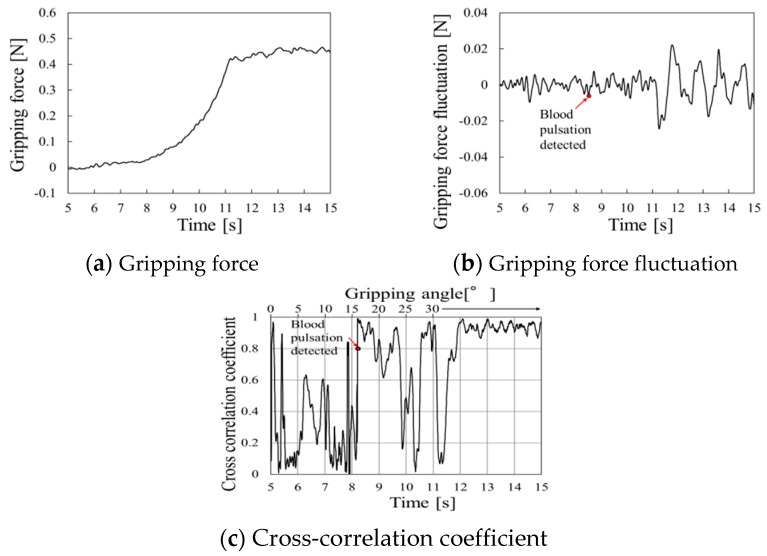
Gripping force behavior for urethane gel at a gripping speed of 5.0°/s.

**Figure 16 sensors-19-05157-f016:**
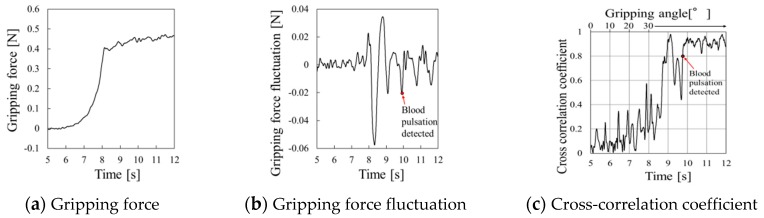
Gripping force behavior for urethane gel at a gripping speed of 10.0°/s.

**Figure 17 sensors-19-05157-f017:**
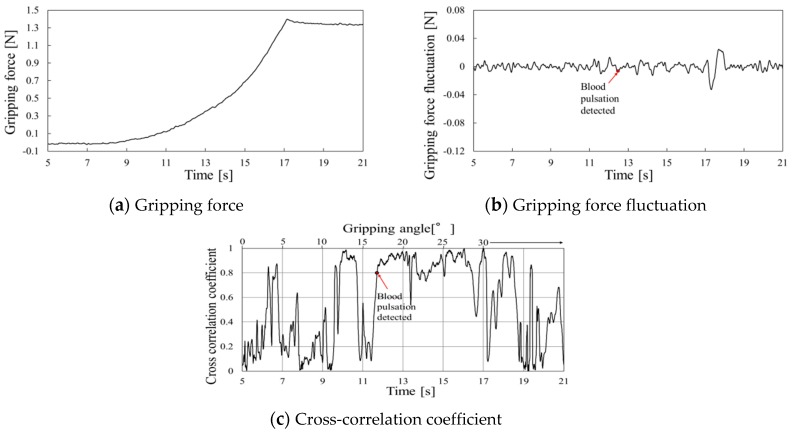
Gripping force behavior for silicone rubber at a gripping speed of 2.5°/s.

**Figure 18 sensors-19-05157-f018:**
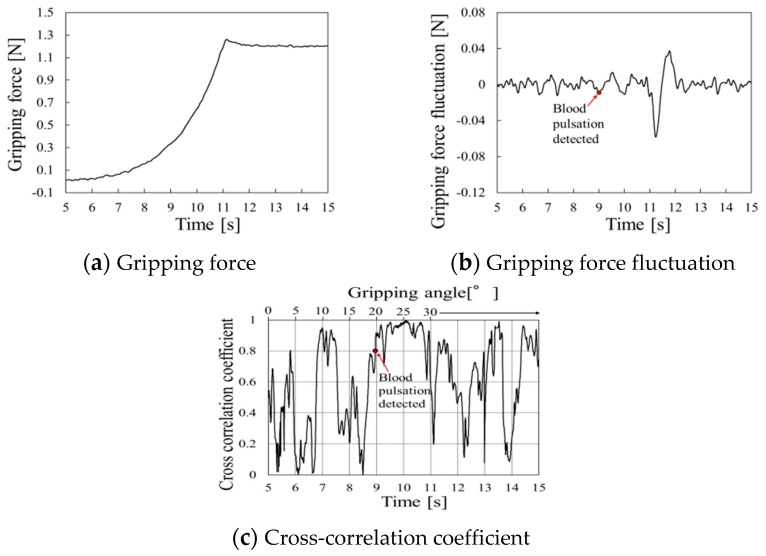
Gripping force behavior for silicone rubber at a gripping speed of 5.0°/s.

**Figure 19 sensors-19-05157-f019:**
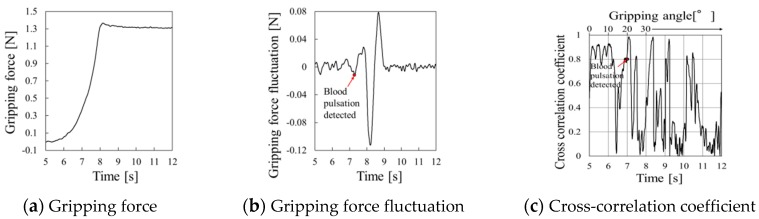
Gripping force behavior for silicone rubber at a gripping speed of 10.0°/s.

**Table 1 sensors-19-05157-t001:** Result of paired t-tests.

Object of T-Test (N = 3)	Gripping Angle [°]	Urethane gel	Silicone Rubber
“Pulse pressure” and “Baseline”	5	P = 0.25	*P < 0.05
	10	P = 0.87	P = 0.14
	15	P = 0.45	*P < 0.05
	20	P = 0.48	*P < 0.05
	25	P = 0.76	*P < 0.05
	30	*P < 0.05	P = 0.23

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
