# Peer review of "Method for the Detection of Tumor Blood Vessels in Neurosurgery Using a Gripping Force Feedback System"

_sensors, 2019, doi:10.3390/s19235157_

Round 1

Reviewer 1 Report

General Comments

The authors report the development of a technique for detecting vessel structures in brain tumour during gripping for robotic neurosurgery applications. During experiments with a synthetic tumor mannequin the authors tested two different schemes for detecting blood pulse based on two proposed metrics as discriminants: the gripping angle of the forceps and the characterisation of the blood pulse waveform, through a cross-correlation coefficient between the measured gripping force, and a gripping estimation from a second-order fluctuation model of the force due to blood pulse.

The mechanical approach is interesting for blood detection in the tumour during robotic surgery in future, however even if the instrumentation is interesting, from sensing processing techniques, the proposed metrics for detection suffer from weak aspects that limit its practical application. Moreover, the paper is confusion in the way is presenting the data analysis and results. Such aspects must be addressed by the authors before the acceptance of the article for publication. Additionally, the manuscripts present some issues to review, regarding formalism, and in-depth review of the English language is required.

Major Comments

The authors claimed that the detection approach is based on the characterisation of the measured blood pulse through a piezoelectric sensor set up at the joint of the end-effector of the gripper. Then, after experiments with two tumor model made from different materials, they presented a valid metric for detection of blood pulse based on the gripping angle, indirectly as a function of the observed gripping force, arguing that statistically at 15 and 30 degrees (for silicone and urethane gel) the approach is able to detect blood pulse. According to the authors' opinion, using the detected force directly is unsuitable as an indictor of blood pulse, due to the non-stationary nature of the fluctuation. However, the use of the characterised gripping angle might be only valid for the specific tested configuration. Some questions may arise. To mention, it is still valid for gripping gestures with the surgical tools manipulating the tissue at an incident axis different than normal axis as tested in the presented experiments. What happens with tumours with different shapes, sizes, of morphologies? The given gripping angles correspond to the gripping plane perpendicular to the tumour axis, but what if the forceps rotate around its axial axis? Summing up, more experimentation with closer to reality conditions must be needed to clarify these questions. How to extrapolate the current results? Following with discussion above, in fact in Table 1, authors present the results of statistic T-Tests evaluating the gripping angle (independent variable) as metric for detecting blood pulse, but the table did not show even the observed gripping force (dependent variable). Is essential to know at least the descriptive statistics of blood pulse for discriminating between “normal” and “pulse pressure”. In the case of the second proposed metric, the cross-correlation coefficient between the measured and the estimated forces, maybe only considered as a value for assessing the agreement between the measured forces and the model estimations, at least in the way as was presented by authors. In this sense, graphs and plots in Figures 13 and 14 reveal a good agreement between the second-order estimation function and the observed forces at variable gripping angles (again, as in point 1, only under the limitation of the assumed experimental condition), but not valid as discriminating criteria. All plots show a coefficient bigger than 0.8, indicative of good agreement. What is needed is again, the statistical descriptors of the significantly different levels of coefficients, and their corresponding force thresholds. No statistics were included. Even if in the Discussion section, authors claimed that the coefficient is useful as a discriminant for detecting blood pulses in real-time, no clear evidence is presented in the paper in the form of quantified correlation coefficient thresholds and statistics. Summing up, a more proper discriminant analysis would be needed, for example in the form of sensitivity-specificity analysis using ROC curves (receiver operating characteristics) or Confusion Matrix, indicating the number of correct detections plus false positives and false negatives rate information to validate the proposed method. Page 5, the paragraph from lines 174-177, where starts as “The waveform…” is confusing. Please rephrase it or clarify the idea. A more in-depth English language review is needed in British English for this journal.

Minor comments

Figure 2, page 2, the label of the subfigure must say “Constant gripping angle.” Instead of “Constant gripping angel”. Page 4, line 122. The statement “… with little delay in real-time processing” is vague. What is understood as “little delay”. Please clarify. Page 4, line 137. Please clarify better what the authors mean for “temperature compensation”. Page 4, line 142, should say something like “Blood vessel pulsation detected as a function of the gripping force”. Page 5, line 156, again, which quantification is deserved for “lower time delay” in the scope of the current experiments? Page 21, line 162-162, “data number” should refer to “data index”, and the “latest data” to “frame data”. Following the line 164, where it says “How many data numbers are set…” is confusing because is written as question form, but more importantly, this should be related to Nyquist criteria for sampling. Please clarify. Page 5, line 169. Where it says, “past 200 data points”, should refer to an “epoch of 200 samplings”. The same for the rest of the paper. Page 10, Section 4.2 should be named as “Test results” as on page 14 for the coherence of the manuscript. Page 11, line 307. Is not needed to repeat “a difference of 5% was found…”. Is clear in statistics and mentioned it once is enough. From page 12, replace please “chapter” by “section”, since properly is a manuscript for a journal, not a book. Page 14, Section 4.2. All paragraph stating “Figure 13 shows the experimental results…” do not mention clearly which kind of results are referred. Is mentioned as the reader continues ahead of the section to notice what the authors indicate. Please clarify in this paragraph. In all section 5.2, where the authors mean “dense”, it should be specified as “data clustering” or something similar. Please, correct. Page 14. In the statement where says “and is continuously plotted in certain angle section in one experiment results”, refers to Figures 13 and 14. As mentioned above, there is no need to include so many plots, because is clear that the correlation coefficient (cc) in all cases is higher than 0.8 which indicates that there is a good agreement between the gripping forces measured and the pulse estimation model, but this does not imply that the selected threshold for cc is adequate as detection discriminant. No statistics about observed statistical descriptors between groups for cc has been reported in all section 5.2. During all the manuscript where the authors refer as “Normal” should be changed for something like “Baseline”, because normal may be interpreted as that the measurements follow a normal distribution, that at sight it seems what is occurs but is not what the authors want to mean. In figures 13 and 14 is better to use different shapes in the bullets instead of colour because for the printed version of the manuscript is difficult to understand the plots. Page 18, lines 466-469, in the statement starting as “As the gripping speed increases…the detection accuracy decreases”. Again, there is no clear quantitative justification for this asseveration, and in any case, the velocity of the gripping at variable speeds is sufficiently slow (at most 10 deg/s) for real-time processing and detection.

Reviewer 2 Report

This paper presents a blood vessel detection method for a master–slave surgical robot system with a force sensor in the slave gripper, and two types of blood vessel detection systems are proposed. The authors conducted several experiments to prove the feasibility of the blood vessel detection method for neurosurgery.

The authors summarize previous studies, but their shortcomings are not summarized. Therefore, it should be added to the introduction section about what difficulties and what research needs are derived from the previous studies. The authors measured grasping force with a sensor-integrated gripper that was developed earlier, but it is not known how the vessel was sensed by the gripper. It is recommended to add if the sensor has specifications such as force accuracy/resolution, sampling rate, and so on to detect blood vessels. In Figure 10(a), the power spectrum of pulse pressure was suddenly high at 30 degrees, and I wonder why. And, further experiments with angles from 25 degrees to 30 degrees can help to determine which trend the power spectrum has increased. Similarly, in Figure 10(b), it is recommended to add an analysis for sudden changes at 30 degrees. Figures 13 and 14 contain a lot of information, but the graphs are hard to understand. It is recommended to add something like analytical pictures or tables to the graph to help you understand it.

Round 2

Reviewer 1 Report

The authors attend the reviewer comments, giving enough responses to most of the issues. At the end the technological set, up presented combines a few features that may be useful for further research. It is not a great breakthrough, but it has value. Because this reviewer still consider that the authors would have the opportunity of significantly enhancing their results, by proposing a more interesting signal detection approach with better and more advanced techniques, but finally the presented material has value and presents enough arguments for publication.

Still a comment about attending. For all the figures 14 to 19, please use the same scale, size and format for all the graphs, to facilitate the interpretation to the reader. Also clearly includes a graphical cue of the precise instant when the blood pulsation detection oscillation is started, also, in the text of all the figure legends. And importantly, enhance the discussion of the observed differences in the stability of the proposed cross-correlation index, depending on the gripping velocity because has been very briefly covered in the Discussion. Now for this reviewer is more evident after the review, but the interpretation maybe not as straightforward for the readers.

Finally, an exhaustive review of the English language is still required before its final acceptance for publication.

Author Response

We wish to express our appreciation to reviewers for their insightful comment. We improved the text and expression of test results in Section 5. Also. In section 2.5, we added a sentence on lines 228-230 and changed Figure 5. In addition, the content in Section 6 was improved. We reanalyzed the results of Section 5 experiments. The results of the experiment were expressed using a better analytical method than before.

Response to Reviewer 1 comment

Point 1: For all the figures 14 to 19, please use the same scale, size and format for all the graphs, to facilitate the interpretation to the reader. Also clearly includes a graphical cue of the precise instant when the blood pulsation detection oscillation is started, also, in the text of all the figure legends.

Response 1: We revised the figures according to the comment. Also, we add the figures to graphical cue of the precise instant when the blood pulsation detection oscillation is started.

Point 2: And importantly, enhance the discussion of the observed differences in the stability of the proposed cross-correlation index, depending on the gripping velocity because has been very briefly covered in the Discussion. Now for this reviewer is more evident after the review, but the interpretation maybe not as straightforward for the readers.

Response 2: please refer to lines 435-443 in the manuscript.

Point 3: Finally, an exhaustive review of the English language is still required before its final acceptance for publication.

Response 3: We once again offered the proofreading of the manuscript to the English proofreader and corrected the English expression.We attached the certificate of English editing service.

Reviewer 2 Report

The reviewer was satisfied by the revision.

Author Response

We wish to express our appreciation to reviewers for their insightful comment. We improved the text and expression of test results in Section 5. Also. In section 2.5, we added a sentence on lines 228-230 and changed Figure 5. In addition, the content in Section 6 was improved. We reanalyzed the results of Section 5 experiments. The results of the experiment were expressed using a better analytical method than before.

We attached the certificate of English editing service.
